# Hierarchical Analysis: Monotonicity of Layer Performance in Large Language Models

## Abstract

We introduce a quantitative framework to evaluate how Large Language Models (LLMs) learn tasks across all layers, revealing a 'monotonicity phenomenon'. Specifically: i) performance at each layer consistently improves from one layer to the next on the pre-training set, and ii) this improvement is consistently observed across various downstream tasks. This monotonicity phenomenon indicates that LLMs effectively capture complex hierarchical features across diverse datasets. For example, our study on the abstraction of concepts using linear representations in word embeddings shows that the clarity of these abstractions progressively increases with each layer. Finally, by leveraging this monotonicity, we can significantly reduce inference time and memory requirements by selecting the most appropriate layer, thereby enhancing the efficiency of LLMs in real-world applications.

## 1 Introduction

Large Language Models (LLMs) have achieved remarkable success across a wide range of domains (Brown, 2020; Bubeck et al., 2023; Chowdhery et al., 2023). Nevertheless, they face significant challenges related to computational and memory demands during inference. To address these challenges and enhance inference efficiency, various techniques have been proposed, such as quantization (Liu et al., 2023), pruning (Sun et al., 2023), and weight sparsification (Frantar & Alistarh, 2023). In addition to these efficiency concerns, LLMs are frequently criticized for their "black box" nature, which has led to numerous studies (Park et al., 2023; Tigges et al., 2023) aiming to investigate and shed light on the underlying mechanisms of how these models function.

Embedding learning plays a crucial role in understanding why LLMs are effective (Park et al., 2023; Tigges et al., 2023; Hernandez et al., 2023; Yan et al., 2024). For instance, (Yan et al., 2024) demonstrated a sequential development pattern in which cognitive abilities are primarily established during pretraining, while expressive abilities are mainly refined through supervised fine-tuning (SFT) and reinforcement learning with human feedback (RLHF). Additionally, embedding learning has been successfully applied across various neural networks with diverse architectures (Kim et al., 2018; Yang & Hu, 2020; Vyas et al., 2024).

In the context of embedding learning in LLMs, the Linear Representation Hypothesis is a crucial concept, suggesting that high-level ideas are represented linearly within the model's representation space. This linear representation indicates that LLMs have the capability to comprehend textual meaning. For example, (Tigges et al., 2023) showed that sentiment is represented linearly in the output layers of LLMs, where a single direction in activation space effectively captures this feature across multiple tasks, with one end corresponding to positive sentiment and the other to negative.

The output from the last layer is typically used as the embedding for downstream tasks (Devlin, 2018; Dosovitskiy, 2020; Radford et al., 2021). However, for a given downstream task, it's worth questioning whether the last layer output of pre-trained models might be overly comprehensive, and whether the outputs from the middle layers could be sufficient as the embedding for that task. Similar to CNNs, where earlier layers capture fundamental visual features like lines and curves, and later layers extract more abstract representations, it is worth exploring whether each layer of pre-trained models could serve as an effective embedding for various tasks.

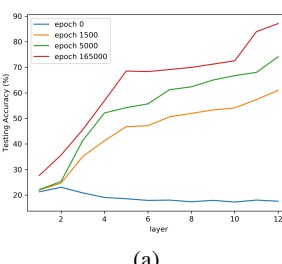 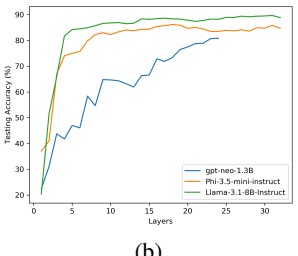 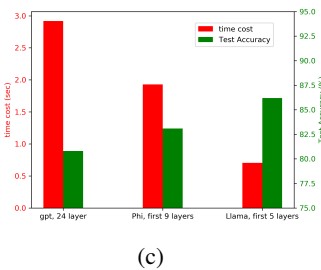

(a)                          (b)                          (c)

Figure 1: Monotonicity and Its Applications: (a) Layerwise accuracy of GPT-2 at different stages of training, using the OpenWebText dataset (Gokaslan & Cohen, 2019); (b) Demonstration of the monotonicity phenomenon in three pre-trained LLMs on the Banking77 dataset; (c) Comparison of the performance of the three LLMs after selecting the most suitable layer embedding on Banking77 dataset.

## 1.1 CONTRIBUTIONS

In this paper, we conduct a hierarchical analysis of LLM performance. We introduce a quantitative and precise characterization of how LLMs learn tasks across all layers by defining the concept of "layerwise performance". Our contribution is as follows:

- Our results show that layerwise performance on pre-trained data progressively improves from one layer to the next, a phenomenon we refer to as "monotonicity". Notably, this phenomenon is absent at the initialization stage of LLMs but becomes increasingly evident as training progresses, as illustrated in Figure 1 (a);

- We then show that for pre-trained LLMs, monotonicity is not limited to the pre-training datasets but also extends to other datasets, indicating that layerwise performance improves progressively across layers even when evaluated on new data. This observation indicates that the ability of LLMs to capture and refine complex features is not limited to the data they were initially trained on but is a more generalizable characteristic. Our experiments provide strong evidence of monotonicity in several widely used LLMs, including Llama 3.1 Instruct (Dubey et al., 2024), Phi-3.5 (Abdin et al., 2024), and GPT-Neo (Black et al., 2022), when tested on diverse datasets such as Banking77 (Casanueva et al., 2020), AG News (Zhang et al., 2015), and the Twitter Sentiment Analysis dataset (Kharde et al., 2016), as illustrated in Figures 1 (b) and 2. To further understand this consistent pattern, we utilize the concept of linear representation, which helps explain why monotonicity reliably emerges across different datasets and model architectures. This analysis not only highlights the robustness and adaptability of monotonicity within LLMs but also offers deeper insights into how these models maintain their performance across various tasks and data domains;

- The concept of monotonicity offers a practical approach to reduce inference time and memory requirements by allowing us to select the most appropriate layer embedding for a given task. When applying this idea to a specific downstream task, we begin by assessing the layerwise performance of the LLMs. If the layerwise performance gains in the final layers start to plateau, it indicates that the model's capacity is likely sufficient for the task. In this scenario, we can identify the layer at the inflection point—where performance gains become marginal—and use this as the cutoff for inference. For instance, as illustrated in Figure 1 (b), the optimal layers are found to be the 5th for Llama 3.1 and the 9th for Phi-3.5. By limiting inference to only the necessary preceding layers, as shown in Figure 1 (c) and 4, we can substantially accelerate the inference process without sacrificing performance. This selective approach enables a more efficient use of LLMs, particularly for resource-constrained applications.

## 1.2 RELATED WORKS

**Hierarchical analysis**  Exploring the intermediate layers of neural networks and LLMs is essential for gaining a deeper understanding of these models and identifying potential issues (Alain, 2016; He & Su, 2024). (Alain, 2016) investigated how features evolve across layers in neural networks by applying linear classifiers independently to each layer's features, revealing that the linear separability of features increases monotonically with depth, highlighting the importance of deeper layers in enhancing the model's ability to differentiate data points. In a similar vein, (He & Su, 2024) introduced a quantitative law describing how contextualized token embeddings are learned through intermediate layers in pre-trained LLMs for next-token prediction, a task relevant to our context. Their findings show that prediction accuracy consistently improves from one layer to the next, confirming our main result.

**Linear representation in LLMs**  The concept of linear representation in LLMs pertains to the idea that high-level features can be expressed in a linear manner, a topic that has been explored in several previous studies (Tigges et al., 2023; Park et al., 2023; Jiang et al., 2024). (Park et al., 2023) formalized this notion by examining how high-level concepts are linearly represented in LLMs, demonstrating that these representations are closely linked to interpretability and control through a non-Euclidean inner product that aligns with the underlying language structure. In a complementary study, (Jiang et al., 2024) investigated the origins of linear representations in LLMs, revealing that both the softmax objective and the implicit bias introduced by gradient descent promote the formation of linear structures in concept representations, as demonstrated through a latent variable model.

**Layer pruning**  Several studies have explored techniques to streamline large language models by addressing layer redundancy and optimizing computation. Research on layer pruning focuses on identifying and removing redundant or unimportant layers, thereby reducing the number of parameters and accelerating inference without significant performance degradationSong et al. (2024); Kim et al. (2024); Chen et al. (2024); Men et al. (2024). Additionally, LayerSkip Elhoushi et al. (2024) introduces a specialized training method that enables the use of only the first half of the model's layers while maintaining high performance. These works collectively highlight the importance of addressing redundancy in LLMs to improve efficiency.

## 2 MONOTONICITY IN LLM

Given a sequence of tokens as input, LLMs function as nonlinear models that iteratively transform the token embeddings into new sequences at each layer using attention mechanisms and other operations.

### 2.1 LAYERWISE PERFORMANCE ON PRE-TRAINED DATA

Consider an LLM composed of $L$ transformer layers, with the pre-training data consisting of $S$ input sequences. Each input sequence, denoted as $X_s$ for $1 \leq s \leq S$, is composed of $T_s$ tokens. For each token position $1 \leq t \leq T_s$, let $x_t^s$ represent the $t$-th token within the sequence $X_s$, expressed as a one-hot encoded vector to capture the token's identity in a high-dimensional space.

During the pre-training phase, the LLM is tasked with learning to predict the next token $x_{T_s}^s$ based on the sequence of preceding tokens $x_1^s, x_2^s, ..., x_{T_s-1}^s$. This predictive process enables the model to gradually capture complex patterns and dependencies inherent in the data. As the input sequence progresses through each of the $L$ transformer layers, richer and more abstract representations of the tokens are formed.

For any given layer $1 \leq l \leq L$, let $h_l^s$ denote the embedding corresponding to the final token in the $l$-th layer for the sequence $x_1^s, x_2^s, ..., x_{T_s-1}^s$. This embedding serves as a refined representation of the input sequence after processing through the first $l$ layers. Consequently, this iterative embedding process across all layers results in the formation of a series of datasets $\{\mathcal{D}_l\}_{l=1}^L$, where each dataset $\mathcal{D}_l$ is defined as $\mathcal{D}_l = \{(h_l^s, x_{T_s}^s) \mid 1 \leq s \leq S\}$.

These datasets collectively represent $L$ distinct classification tasks, one for each layer, effectively capturing how the model's ability to encode and predict token information evolves as it progresses through the layers. This layered structure offers insights into the intermediate representations learned by the LLM during the pre-training process, highlighting how each layer contributes to refining the model's understanding of sequential data.

Given the $L$ distinct datasets $\{\mathcal{D}_l\}_{l=1}^{L}$ generated from the pre-trained LLM, we can evaluate the performance of each layer of the LLM through the following procedure. First, we split each dataset $\mathcal{D}_l$ into a training set and a testing set to facilitate model evaluation. For each layer $l$, we then apply a logistic regression model to the training set of dataset $\mathcal{D}_l$ to learn the relationship between the layer's embedding and the corresponding target tokens.

Once the logistic regression model is trained on this training set, we evaluate its predictive accuracy using the testing set. The accuracy obtained from this evaluation serves as a quantitative measure of the **layerwise performance**, reflecting how well the embeddings produced by the $l$-th layer of the LLM capture the necessary information for predicting the next token. By repeating this process for all $L$ layers, we gain a comprehensive understanding of how the predictive capability of the LLM evolves across different layers, providing insights into the effectiveness of each layer's representations in contributing to the overall learning process.

## 2.2 MONOTONICITY OF LAYERWISE PERFORMANCE IN GPT-2

To investigate the layerwise performance in LLMs, we conducted an analysis using the widely adopted GPT-2 model (Radford et al., 2019) in conjunction with the OpenWebText dataset. Specifically, we trained the small variant of GPT-2 using this dataset to observe how layerwise performance evolves over time during the training process. At various training stages, specifically at epochs 0, 1,500, 5,000, and 165,000, we recorded the performance of each layer to gain insights into the model's learning progression, presented in Figure 1 (a).

Figure 1 (a) illustrates the layer-wise performance of a pre-trained GPT-2 model, demonstrating a clear increase in testing accuracy across layers, particularly at epoch 165,000. This trend, where the accuracy progressively improves from the lower layers to the upper layers, is referred to as monotonicity. As the layers deepen, the model's ability to capture and represent complex patterns in the data becomes more pronounced, leading to higher accuracy rates. This phenomenon suggests that deeper layers in the fully trained GPT-2 model contribute more significantly to the model's overall predictive capability, reinforcing the importance of multi-layered architectures in LLMs like GPT-2.

Furthermore, Figure 1 (a) captures the evolution of this monotonicity throughout the training process. Notably, at the initialization stage (epoch 0), the model exhibits no clear monotonic trend, with the testing accuracy remaining relatively low and flat across all layers. As training progresses, however, monotonicity becomes increasingly evident, with significant gains in layerwise performance emerging by epochs 1,500, 5,000, and ultimately, 165,000. This progressive emergence of monotonicity underscores how the training process enables the model to develop a more refined understanding of language, leading to enhanced representation and predictive capabilities as training deepens. This observation highlights that monotonicity is not an inherent characteristic of LLMs at initialization but rather a phenomenon that emerges and strengthens as the model learns from data over time.

The concept of monotonicity has been explored in prior research (Alain, 2016; He & Su, 2024). For instance, He et al. (2024) applied a least squares fit on the dataset $\mathcal{D}_l$ and introduced the concept of the prediction residual (PR) to measure the LLM's capability for next-token prediction, with their findings closely aligning with our results. Moreover, the phenomenon of monotonicity is not limited to LLMs; it can also be observed in neural networks with different architectures, such as FCNs and CNNs (Alain, 2016). This suggests that monotonicity is a fundamental characteristic of hierarchical network models trained by gradient descent algorithm.

## 3 MONOTONICITY ACROSS DATASETS

The previous section demonstrated that monotonicity is present in pre-trained data and becomes progressively more pronounced as training advances. Given that pre-trained LLMs are widely favored for their strong generalization abilities on downstream tasks and datasets, it raises an intriguing question: **Does monotonicity persist when the pre-trained model is applied to new tasks?** Addressing this question is crucial, as it can provide valuable insights into the behavior and adaptability of LLMs, informing their effective application across a broader range of real-world tasks.

### 3.1 LAYERWISE PERFORMANCE ACROSS DATASETS

Consider a pre-trained LLM with $L$ transformer layers, and a new dataset consisting of $S$ input sequences $\{X_s\}_{s=1}^S$ along with their corresponding labels $\{Y_s\}_{s=1}^S$. For any layer $1 \leq l \leq L$, let $h_l^s$ denote the embedding of the final token in the $l$-th layer for the input sequence $X_s$. This embedding serves as a progressively refined representation of the input sequence as it is processed through the first $l$ layers of the LLM. Through this process, we construct a series of datasets $\{\mathcal{D}_l\}_{l=1}^L$, where each dataset $\mathcal{D}_l$ is defined as $\mathcal{D}_l = \{(h_l^s, Y_s) \mid 1 \leq s \leq S\}$.

Similarly, to evaluate the performance of each layer, we further divide each dataset $\mathcal{D}_l$ into separate training and testing sets. For each layer $l$, a linear model—either logistic regression for discrete labels or linear regression for continuous labels—is then fitted to the training set of dataset $\mathcal{D}_l$ and evaluates the linear model on the testing set, which is considered as the **layerwise performance across the dataset**. This approach allows us to learn and assess the relationship between the layer's embeddings and the corresponding labels, providing insights into how well the representations at each layer capture the relevant features for the given task.

### 3.2 MONOTONICITY ACROSS DATASETS

To investigate the layerwise performance across datasets, we conducted an analysis using some famous LLMs including Llama 3.1 Instruct, Phi-3.5 and GPT-Neo, and in conjunction with some NLP datasets including AG News, Banking77 and Twitter Sentiment Analysis dataset. The results is presented in Figure 2.

Figure 2 demonstrates that all three pre-trained LLMs—Llama 3.1 Instruct, Phi-3.5, and GPT-Neo—exhibit the phenomenon of monotonicity across the three NLP datasets: AG News, Banking77, and the Twitter Sentiment Analysis dataset, indicating their ability to generalize effectively to new tasks. monotonicity, which we first observed in pre-training data, is characterized by the progressive improvement in accuracy as data passes through successive layers of the model. The fact that this trend continues when the LLMs are applied to entirely new datasets suggests that the representational power of these models, acquired during pre-training, transfers well to different downstream tasks. This generalization ability is crucial because it highlights the adaptability of LLMs to learn complex features even outside the original training domain, confirming that their deeper layers can refine and enhance understanding across various contexts and tasks, beyond what was learned during pre-training.

When examining the degree of monotonicity for a given pre-trained LLM, it becomes evident that this phenomenon varies in intensity. For instance, in some cases, the performance improvement across layers is relatively flat in the last few layers, while in others, it is steep. A flatter monotonicity curve may imply that the model's current size is sufficient for effectively capturing the information needed for the task, suggesting that adding more layers might yield diminishing returns in terms of performance gains. Conversely, a steeper monotonicity curve suggests that the model is still gaining significant representational power as layers are added, indicating that the current model size might not be large enough to fully capture the complexity of the task at hand. Therefore, the shape of the monotonicity curve provides insights into whether a model has reached its optimal capacity or if there is room for further improvement with additional layers.

When comparing monotonicity across different models on the same dataset, it is apparent that each LLM exhibits unique learning capabilities. For instance, the Llama 3.1 Instruct model shows a different pattern of layerwise performance compared to Phi-3.5 and GPT-Neo, reflecting their varying abilities to extract and understand the semantic meaning of sentences within each dataset. These dif-

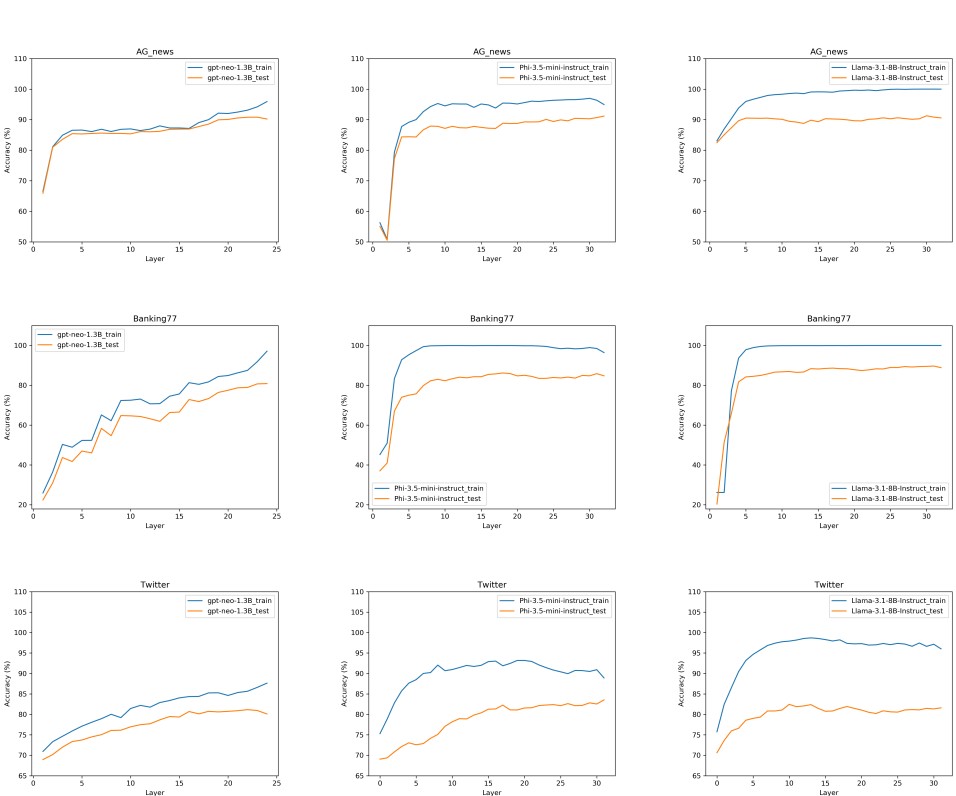

Figure 2: Monotonicity Across Datasets. The x-axis corresponds to the layer index, while the y-axis represents the accuracy for both training and testing sets. Monotonicity is consistently present across the various new tasks. However, this phenomenon is not uniform; it differs based on the interaction between each dataset and the specific pre-trained LLM being used.

ferences in monotonicity indicate that some models are more adept at learning the relevant features for a given task, while others may require more layers or different architectures or even different training processes to achieve similar levels of performance. Thus, Figure 2 not only highlights the existence of monotonicity but also underscores how this phenomenon varies depending on the model and inherent ability to generalize across different NLP tasks.

### 3.3 EXPLANATION OF MONOTONICITY ACROSS DATASETS

**Linear representation** The concept of linear representation in LLMs refers to the idea that high-level features and relationships between words can be captured and expressed in a linear fashion, an area that has been investigated in several studies (Tigges et al., 2023; Park et al., 2023; Jiang et al., 2024). A classic example of this phenomenon is found in word embeddings, where it has been empirically observed that pairs like embedding("woman") - embedding("man") and embedding("queen") - embedding("king") are nearly parallel in the vector space, indicating that the model has effectively captured the underlying semantic relationship between these words (Mikolov et al., 2013). This parallelism suggests that the embedding space represents consistent linear relationships that encode concepts such as word inflection, gender or other analogies.

In our study, we applied the idea of linear representation to LLMs by examining whether similar semantic pairs maintain such linear relationships across different layers of the model. For each pair of related words, such as word A and word B, we extracted their embeddings from each layer of the LLM and computed the vector difference between them. We then calculated the inner product of these differences across all similar word pairs. If the resulting inner product is close to 1, this indicates that the embeddings effectively capture the intended concept, contributing to the model's ability to understand the meanings of sentences. We evaluated this phenomenon in two models, GPT-Neo and Llama 3.1 Instruct, across all layers, with the results presented in Figure 3.

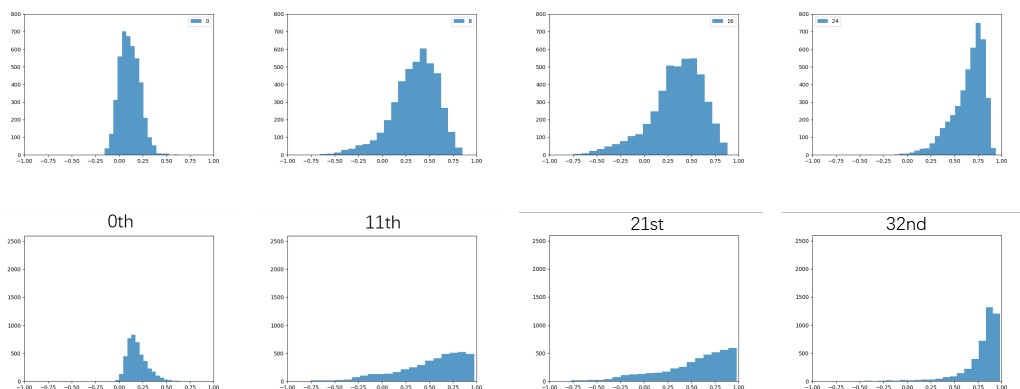

Figure 3: Linear Representation in GPT-Neo and Llama 3.1: The 0th layer represents the input embedding, which transforms the one-hot encoded input into continuous, lower-dimensional vectors. The first row of the figures corresponds to GPT-Neo, while the second row pertains to Llama 3.1. The impact of linear representation becomes increasingly evident in both GPT-Neo and Llama 3.1. This effect is significantly more pronounced in Llama 3.1, where the values approach 1, suggesting that its layerwise embeddings capture and comprehend the text more effectively than those in GPT-Neo. This indicates that Llama 3.1's embeddings are more adept at representing the underlying relationships and concepts within the text as they advance through the layers.

Figure 3 demonstrates that the impact of linear representation becomes increasingly evident as we progress through the layers of LLMs, indicating a greater ability to capture and comprehend the meaning of the text in the deeper layers. This trend suggests that as data traverses through successive layers, the model continuously refines its understanding and representation of the underlying semantics. This refinement process helps to explain why the phenomenon of monotonicity is observed across different new datasets.

Additionally, this effect is notably more pronounced in Llama 3.1, where the values consistently approach 1 across layers, signifying a stronger alignment in capturing semantic relationships. This indicates that Llama 3.1's layerwise embeddings are more adept at understanding and encoding the nuances of the text compared to those of GPT-Neo. Consequently, Llama 3.1 exhibits a superior capability to accurately represent the intended meaning and context when applied to new datasets.

# 4 APPLICATION OF MONOTONICITY

The previous section demonstrated the phenomenon of Monotonicity across various datasets, revealing that it varies depending on the interaction between each dataset and the specific pre-trained LLM. This means that the degree of performance improvement across layers is not uniform. For certain datasets and LLMs, the increase in accuracy flattens in the last few layers. According to the definition of layer performance, this suggests that the embedding from the final layer might be overly sufficient for the datasets, even though it is common practice to use the output from the last layer as the primary embedding for downstream tasks (Devlin, 2018; Dosovitskiy, 2020; Radford et al., 2021). To expedite the inference process with the given dataset and LLM, we can proceed with the following steps:

- **Step 1:** Assess the layer-wise performance of the LLM on this dataset.
- **Step 2:** Identify the layer that produces embeddings comparable in effectiveness to those generated by the final layer.
- **Step 3:** Utilize the embeddings from the identified layer for downstream tasks associated with this dataset.

A more concrete example of this method can be observed with the Banking77 dataset when using Llama 3.1, as shown in Figure 1 (b) and Figure 2. In this case, the performance of the embeddings from the 5th layer is nearly as effective as those from the last layer, indicating that the deeper layers do not always provide significant additional benefits for certain tasks. Therefore, it may be advantageous to use only the first five layers of Llama 3.1 during the inference process, resulting in substantial savings in both time and memory without sacrificing model performance. More detailed examples supporting this observation can be found in Figure 4.

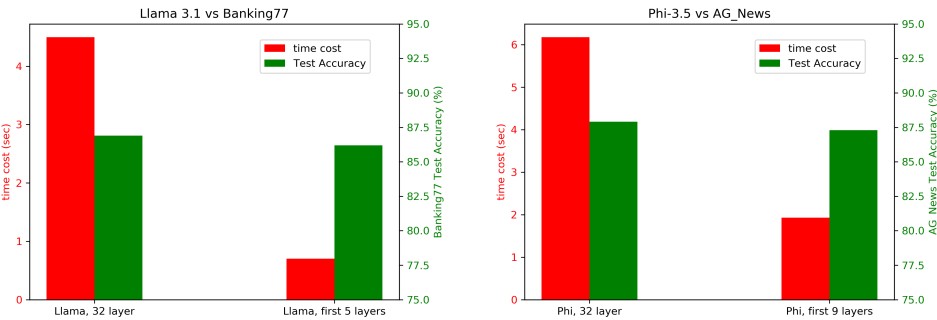

Figure 4: Reducing Inference Time While Maintaining Performance Through Strategic Layer Embedding Selection: By carefully selecting the most appropriate layer embeddings, it can significantly reduce inference time without compromising the model's performance.

Figure 4 demonstrates that this approach significantly optimizes computational efficiency. For instance, in the case of the Banking77 dataset, selecting the appropriate layer embedding from Llama 3.1 can lead to an inference time reduction of nearly 85%, with virtually no loss in performance. Similarly, for the AG News dataset, choosing the optimal layer embedding from Phi-3.5 results in a reduction of inference time by approximately 70%, while still maintaining high accuracy. Furthermore, by leveraging the concept of monotonicity, we can not only identify the most suitable pre-trained LLM for a given downstream task but also select the optimal layer embeddings based

on the specific performance and inference time requirements, as illustrated in Figures 1 (b) and (c). This makes the approach highly adaptable and efficient for various real-world applications.

# 5 Experiments

**Settings in Figure 1 (a)** We pre-trained the GPT-2 small model, which consists of 12 transformer blocks and approximately 124 million parameters, using the OpenWebText dataset. The training was conducted on an 8X A100 80GB node. The process continued until the model's training loss reached approximately 3.15, taking around 4 days to complete. Throughout the training, we recorded the model parameters at specific checkpoints—epochs 0, 1,500, 5,000, and 165,000. For each of these checkpoints, we evaluated the layerwise accuracy on the OpenWebText dataset to assess the model's performance at various stages of training, as presented in Figure 1 (a).

**Settings in Figure 1 (b) and Figure 2** The model sizes used in our experiments are as follows: 1.3 billion parameters for GPT-Neo, 3.8 billion parameters for Phi-3.5-mini-Instruct, and 8 billion parameters for Llama 3.1 Instruct. We conducted the experiments according to the procedure outlined in Section 3. Specifically, the experiments for GPT-Neo and Phi-3.5-mini-Instruct were run on a single RTX 4090 GPU with 24GB of memory, while the experiment for Llama 3.1 Instruct was conducted on an A100 GPU with 40GB of memory.

The corresponding datasets are detailed as follows: For the AG News dataset, we randomly selected 10,000 samples from the training set and used all 7,600 samples from the testing set. This dataset contains four label categories, $Y = \{0, 1, 2, 3\}$. The Banking77 dataset comprises 10,003 training samples and 3,080 testing samples, with a total of 77 label categories, $Y = \{0, 1, ..., 76\}$. Lastly, the Twitter Sentiment Analysis dataset includes 10,223 training samples and 4,382 testing samples, with two label categories, $Y = \{0, 1\}$.

**Settings in Figure 3** We utilized the word analogy dataset introduced by (Gladkova et al., 2016) and specifically selected the "noun - plural" analogy category to conduct the linear representation experiment. The experiments were carried out following the procedure outlined in Section 3. For the GPT-Neo model, the experiment was run on a single RTX 4090 GPU with 24GB of memory, while the Llama 3.1 Instruct experiment was conducted on an A100 GPU with 40GB of memory.

**Settings in Figure 1 (c) and Figure 4** To measure the inference time, we randomly selected the same 10 sentences from each corresponding dataset. The testing accuracy was then calculated using the entire testing set for each dataset.

# 6 Discussion

In this paper, we presented a detailed hierarchical analysis of LLMs, uncovering the phenomenon of monotonicity in layerwise performance. Our study reveals that this progressive improvement in accuracy, observed across successive layers, is not only evident in pre-training data but also extends to a variety of downstream tasks. This demonstrates the ability of LLMs to adapt and generalize to new datasets, making them highly versatile across different applications. We introduced the concept of linear representation to provide further insights into why monotonicity emerges, highlighting the structured way in which LLMs capture and refine complex features. By identifying that layerwise performance can plateau in the later stages, we also showed how this understanding can be harnessed to optimize inference time and memory usage. Selecting the optimal layer embeddings for specific tasks enables significant efficiency gains without compromising model performance, offering a practical solution for deploying LLMs in resource-constrained environments.

However, a limitation of our study is that it does not fully address how the interaction between the data and the pre-trained LLMs influences monotonicity. For instance, if a dataset contains random labels without any inherent structure, the phenomenon of monotonicity is unlikely to emerge, as there is no meaningful relationship for the model to learn. Similarly, monotonicity may not manifest when the dataset is unrelated to the LLM's pre-training domain, such as using non-NLP datasets with LLMs that were pre-trained on language tasks. These observations suggest that monotonicity is not a universal property but rather depends on the alignment between the dataset characteristics

and the LLM's learned representations. Future work should explore these interactions more deeply to better understand the conditions under which monotonicity emerges, thereby providing a more comprehensive framework for applying LLMs to a broader range of tasks.

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
