# A APPENDIX

In this section, we conduct more experiments to explore more metrics and datasets to check the monotonicity.

**Perplexity in GPT-2**   We evaluated the performance of GPT-2 using Perplexity on the OpenWeb-Text dataset in different epochs. The Perplexity is calculated as follows:

$$\text{Perplexity} = \exp\{-\frac{1}{N}\sum_{i=1}^{N}\log p_\theta(w_i|w_1,...,w_{i-1})\} \tag{1}$$

where $p_\theta(w_i|w_1,...,w_{i-1})\}$ is the probability assigned by the model to the $i$-th word, given the preceding words $w_1,...,w_{i-1}$ and $N$ is the total number of words in the sequence. The results are shown as follows: The results demonstrate that the monotonicity principle remains effective across

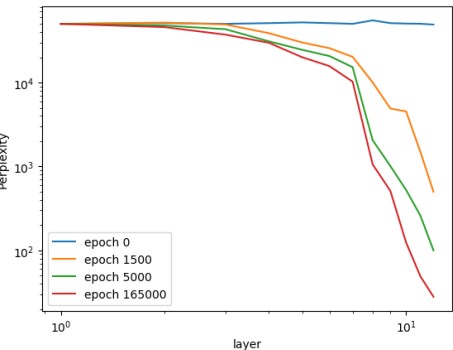

Figure 5: Monotonicity of Perplexity in GPT-2 during training process

different metrics, even after the training process.

**Log-Loss for Classification**   To assess classification performance, we calculated the log-loss on the Banking77 dataset across three models (GPT-neo, Phi-3.5 and Llama 3.1). The log-loss is defined as follows:

$$\text{Log-loss} = -\frac{1}{N}\sum_{i=1}^{N}\sum_{c=1}^{C}y_{i,c}\log p_{i,c} \tag{2}$$

where $N$ is the total number of words in the sequence and $C$ is the total number of classes. $y_{i,c}$ is a binary indicator (1 or 0) that specifies whether class $c$ is the correct class for sample $i$. $p_{i,c}$ is the predicted probability of sample $i$ belonging to class $c$. The results are shown as follows: The results indicate that the monotonicity property is consistently upheld across these models on this metric as well.

**MMLU Question Answering - Humanities**   We conducted experiments on the MMLU Humanities question-answering benchmark to evaluate two models. The dataset consists of questions with four answer options, where only one option is correct. For instance, a sample input might be: "question: What is the embryological origin of the hyoid bone?. answer: A. The first pharyngeal arch, B. The first and second pharyngeal arches", C. The second pharyngeal arch, D. The second and third pharyngeal arches" and the label is "D". This problem is framed as a four-class classification task. We evaluated the monotonicity of two models, Phi-3.5 and Llama 3.1, with the results presented as follows: The findings confirm that the monotonicity principle remains valid for this task. However, due to the complexity of this dataset, there are no accuracy flattens in the last few layers, meaning the proposed method cannot be directly used to accelerate the inference process in this case.

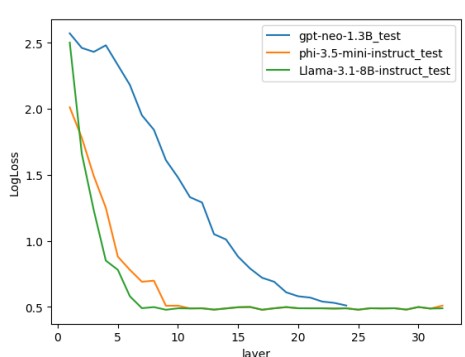

Figure 6: Monotonicity of Log-Loss across three models

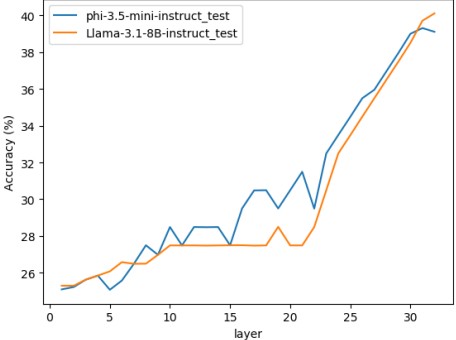

Figure 7: Monotonicity across two models for MMLU Humanities

**Regression Data**  For regression tasks, we evaluated the Yelp dataset for rating prediction ($Y = \{1, 2, 3, 4, 5\}$) using Mean Squared Error (MSE) as the evaluation metric. We randomly select 10000 samples with balanced labels (the sample size of each label is 2000). We use 70% of the sample for training and 30% for testing. We test the monotonicity of three models (GPT-Neo, Phi-3.5 and Llama 3.1) and the results are shown as follows:

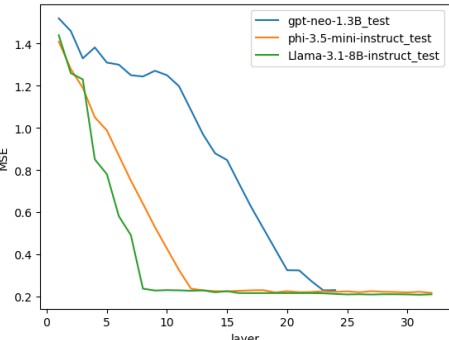

Figure 8: Monotonicity across three models for regression data

The results confirm that the monotonicity property holds consistently for all three models tested on this regression dataset.

**Monotonicity when increasing training samples**  With a smaller dataset, the monotonicity property may not be immediately apparent due to the limited number of samples, which can introduce variability and noise into the results. However, as the sample size increases, the underlying patterns become more pronounced, and the monotonicity becomes more evident, which is presented as follows: The experiment described above utilizes the Banking77 dataset with varying training sample

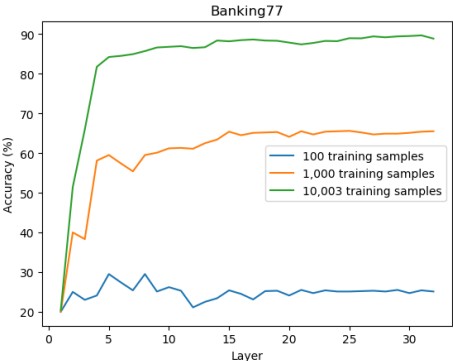

Figure 9: Monotonicity across different training sample sizes

sizes to assess performance. The model used for these evaluations is Llama 3.1 8B.