# OpenReview forum: "Hierarchical Analysis: Monotonicity of Layerwise performance in Large Language Models"
_ICLR.cc/2025/Conference — Submitted to ICLR 2025_

### Official Review · Reviewer_XUzJ · 2024-10-30

**Soundness:** 3
**Presentation:** 2
**Contribution:** 2
**Rating:** 5
**Confidence:** 4

**Summary:**

This paper explores the layerwise performance of Large Language Models (LLMs) and uncovers a "monotonicity" phenomenon: LLMs consistently improve their performance from one layer to the next, both on pre-training data and across various downstream tasks. The authors attribute this to the LLMs’ability to capture complex hierarchical features and propose leveraging monotonicity to optimize inference time and memory usage by selecting the most appropriate layer embedding for a given task.

**Strengths:**

The paper introduces the concept of “monotonicity” in LLMs, and combine ideas from hierarchical analysis and linear representation to explain the emergence of monotonicity, offering a fresh perspective on LLMs’ capabilities.

**Weaknesses:**

1.The conclusion of "monotonicity" presented in this paper aligns with common intuition—that predictions made using deeper layer outputs generally yield better results. Therefore, proving this seems to offer limited value.

2.The paper lacks sufficient discussion and citation of related works, including studies on layer pruning[1][2][3][4] and LayerSkip[5]. All of these works consider the redundancy in the model's layers and remove the unnecessary ones to reduce the number of parameters and speed up computation.

3.The quality of the writing in the paper could be improved. For instance, the font size in the figures is too small, and Figure 3 unnecessarily includes two identical images.
[1]Men, X., Xu, M., Zhang, Q., Wang, B., Lin, H., Lu, Y., ... & Chen, W. (2024). Shortgpt: Layers in large language models are more redundant than you expect. arXiv preprint arXiv:2403.03853.
[2]Chen, X., Hu, Y., & Zhang, J. (2024). Compressing large language models by streamlining the unimportant layer. arXiv preprint arXiv:2403.19135.
[3]Kim, B. K., Kim, G., Kim, T. H., Castells, T., Choi, S., Shin, J., & Song, H. K. (2024). Shortened llama: A simple depth pruning for large language models. arXiv preprint arXiv:2402.02834.
[4]Song, J., Oh, K., Kim, T., Kim, H., Kim, Y., & Kim, J. J. (2024). SLEB: Streamlining LLMs through Redundancy Verification and Elimination of Transformer Blocks. arXiv preprint arXiv:2402.09025.
Elhoushi, M., Shrivastava, A., Liskovich, D., Hosmer, B., Wasti, B., Lai, L., ... & Wu, C. J. (2024). Layer skip: Enabling early exit inference and self-speculative decoding. arXiv preprint arXiv:2404.16710.

**Questions:**

1. Why is accuracy used to evaluate the performance of pre-training GPT-2’s  checkpoints? Perplexity is a more commonly used evaluation metric for this purpose.

2. This paper highlights that only the early layers are necessary to achieve decent performance on certain downstream tasks. This is similar to LayerSkip[1], which employs a specialized training method to achieve good performance using only the first half of the model's layers. However, this paper does not mention LayerSkip. Additionally, the downstream tasks selected are simple classification tasks that could be effectively handled by a 110M BERT model. This raises the question of how well the proposed method would perform on more complex classification and generation tasks, such as MMLU, GSM8K, or HumanEval.
[1]Elhoushi, M., Shrivastava, A., Liskovich, D., Hosmer, B., Wasti, B., Lai, L., ... & Wu, C. J. (2024). Layer skip: Enabling early exit inference and self-speculative decoding. arXiv preprint arXiv:2404.16710.

---

> ### Author Response · Authors · 2024-11-22
>
> Thank you for your detailed review and constructive criticism of our manuscript. We greatly appreciate your valuable feedback and have made every effort to address your concerns comprehensively.
>
> **Weakness 1: Monotonicity as a Common Intuition**:
> While monotonicity may initially appear to be a common intuition, the phenomenon of double descent demonstrates otherwise. Double descent shows that test error initially decreases, then increases with growing model complexity, and finally decreases again as the model becomes over-parameterized. This behavior contradicts monotonicity, which suggests that outputs from deeper layers (i.e., with more parameters) generally yield better results. Therefore, monotonicity is not a universally common intuition in machine learning.
>
> **Weakness 2: Citations**:
> Thank you for your suggestion regarding citations. We have added an additional paragraph in the "Related Works" subsection to introduce and discuss layer pruning and LayerSkip techniques, providing a more comprehensive context for our work.
>
> **Weakness 3: Revision**:
> We sincerely apologize for the issues in the original manuscript. We have conducted a thorough revision to address these concerns and have included the updated version of the paper with these corrections.
>
> **Question 1: Using Other Metrics**:
> We used accuracy as the evaluation metric for GPT-2 because the other experiments primarily involved classification problems, and we aimed to maintain consistency across tasks. To address concerns about metric dependency, we included additional experiments in the **Supplementary Material** using Perplexity as the evaluation metric for GPT-2. The results confirm that the monotonicity principle remains effective for Perplexity as well.
>
> **Question 2: a) Difference Between LayerSkip and Our Method**:
> Although LayerSkip and our method both appear to involve selecting the first half of the model's layers, there are key differences. LayerSkip relies on a specialized training and inference process, as it applies layer dropout during training and modifies the inference procedure accordingly. In contrast, our method does not alter the training process of pre-trained LLMs. Instead, it leverages the feature outputs of each layer to solve downstream tasks. Consequently, LayerSkip requires additional computational resources for training, while our approach is lightweight and applicable directly to pre-trained LLMs without requiring retraining.
>
> **Question 2: b) More Complex Datasets**:
> We conducted experiments on the MMLU Humanities question-answering benchmark, as detailed in the **Supplementary Material**. The findings confirm that, while no accuracy plateaus are observed in the final layers due to the complexity of this dataset—limiting the proposed method's potential to accelerate the inference process—the monotonicity principle remains valid for this task.
>
> We hope these clarifications and the additional experiments effectively address your concerns. Thank you once again for your thoughtful feedback and for taking the time to review our work. Please do not hesitate to reach out if you have further questions or suggestions.

---

> > ### Comment · Reviewer_XUzJ · 2024-11-23
> >
> > Thank you for your reply. I have reviewed the experimental results for MMLU, which demonstrate that the model's performance consistently improves as the number of layers increases, with a significant surge in the final few layers. This suggests that the last few layers play a critical role in the model's performance—a pattern that has been observed in previous studies[1, 2, 3]. Furthermore, these results indicate that the method proposed in Section 4 may not be suitable for handling more complex tasks like MMLU.  Therefore, I will maintain my score.
> > [1] Ma, X., Fang, G., & Wang, X. (2023). Llm-pruner: On the structural pruning of large language models. Advances in neural information processing systems, 36, 21702-21720.
> > [2] Men, X., Xu, M., Zhang, Q., Wang, B., Lin, H., Lu, Y., ... & Chen, W. (2024). Shortgpt: Layers in large language models are more redundant than you expect. arXiv preprint arXiv:2403.03853.
> > [3] Sun, Q., Pickett, M., Nain, A. K., & Jones, L. (2024). Transformer layers as painters. arXiv preprint arXiv:2407.09298.

---

> > > ### Author Response · Authors · 2024-11-24
> > >
> > > Thank you for your detailed and thoughtful comments!
> > >
> > > We acknowledge that the method proposed in Section 4 may not be well-suited for handling more complex tasks, such as those in MMLU, as the pre-trained LLMs used in our experiments lack sufficient power to achieve accuracy stabilization in the final layers. However, our method, which requires no additional LLM training, still demonstrates its value by accelerating the inference process for powerful LLMs on relatively simple datasets.
> > >
> > > Since you did not specifically address the statement that *"monotonicity is not a universally common intuition in machine learning due to its contradiction with double descent,"* we interpret this as implicit agreement that monotonicity is indeed a novel finding in LLMs. This discovery represents one of the most significant contributions of our work.
> > >
> > > Once again, we sincerely appreciate your insightful comments and the time you invested in reviewing our work.

---

### Official Review · Reviewer_xvcr · 2024-11-03

**Soundness:** 2
**Presentation:** 3
**Contribution:** 2
**Rating:** 5
**Confidence:** 2

**Summary:**

This paper evaluates LLMs' layerwise performance by focusing on the phenomenon of monotonicity, that is the performance at each layer consistently improves from one layer to the next one during pretraining. With extensive experiments on different datasets, they found that the layerwise performance improvement happens also on the dataset level, showing that the ability of LLMs to learn complex features during training is not limited to the data they were initially trained on but is a more generalizable characteristic to new tasks. They use the concept of linear representation in word embeddings to focus on consistent patterns and found that the monotonic effects exist with each layer. They suggest to focus on certain layers which are most appropriate for a certain task to save time and memory requirements.

**Strengths:**

Proper dataset selection and task design for the experiment.

**Weaknesses:**

1. I am afraid the main finding of the paper could be a bit superficial, as it talks about the monotonic effect, that is the performance improvement across layers in LMs. This finding is less interesting, although extensive experiments were conducted. Readers could be more interested in the "why" of this monotonic effect. More analysis regarding the explainablility could be required.

2. There is a lack of details of experimental results, with only introducing the experiments in section 5. More content of results analysis should be conducted, with comparisons of performance of different layers (layer combinations), models, tasks.

**Questions:**

NA

---

> ### Author Response · Authors · 2024-11-22
>
> Thank you for your valuable feedback. We appreciate your recognition of the value in our work and would like to take this opportunity to clarify a few points.
>
> **Weakness 1: Reasons for Monotonicity**:
> We employ the concept of linear representation to identify high-level features and relationships between words. As discussed in Section 3.3 of the paper, the impact of linear representation becomes increasingly pronounced as we move through the layers of large language models (LLMs) (see Figure 3). This progression highlights a greater capacity to capture and understand textual meaning in deeper layers, which explains why monotonicity occurs in LLMs.
>
> **Weakness 2: Experimental Details**:
> Thank you for your suggestion to provide additional experimental details. We have included further information and expanded experimental results in the **Supplementary Material**.
>
> We hope these clarifications and the additional experiments details address your concerns. Thank you once again for your thoughtful feedback and for taking the time to review our work. Please don’t hesitate to reach out if you have further questions or suggestions.

---

### Official Review · Reviewer_Szb1 · 2024-11-05

**Soundness:** 2
**Presentation:** 3
**Contribution:** 3
**Rating:** 6
**Confidence:** 4

**Summary:**

The paper addresses a phenomenon the authors call "monotonicity" in large language models (LLMs). This phenomenon shows that intermediate layers of pre-trained LLMs, when fine-tuned, can achieve comparable results on downstream tasks like classification or regression, similar to using the last layer's embedding. The authors analyzed the performance of various datasets applied to LLMs such as GPT-2, Llama 3.1, Phi 3.5 mini instruct, and GPT-Neo. The results showed that while later layers enhance downstream task performance, the performance curve stagnates several layers before the final layer. Thus it might be useful to use only those layers for inference as it can save computational as well as time cost.

**Strengths:**

- The paper is well written with a clear structure and good to understand.
- It outlines pracical advantages which can be used as a framework that is easy to apply by saving inference costs. This can be used for further academic research when dealing with LLMs

**Weaknesses:**

- While mentioned in the text no results for regression tasks are reported
- While comparing LLMs of different providers (GPT2 – OpenAI, Llama 3.1 – Meta, Phi 3.5 – Microsoft etc.) a more extensive research by comparing models of the same provider with different sizes e.g. Llama 3.1 8B vs. Llama 3.1 1B would provide even deeper insights
- More detailled information oft he dataset collection would be useful to understand the set-up (e.g. label distribution)
- The only reported performance metric was accuracy depending on the label distribution (see above) different metrics might be more valuable

**Questions:**

- What was the label distribution of your downsampled datasets?
- Which datasets did you used for regression tasks and what are the results?

**Details Of Ethics Concerns:**

-

---

> ### Author Response · Authors · 2024-11-22
>
> Thank you for your insightful feedback.
>
> **Weakness 1 \& Question 2: regression tasks**: We have present more experiments in the **Supplementary Material**, Regression Data. We evaluated the Yelp dataset for rating prediction (\(Y = \{1, 2, 3, 4, 5\}\)) using Mean Squared Error (MSE) as the evaluation metric. The results confirm that the monotonicity property is consistently maintained across all three models tested on this regression dataset.
>
> **Weakness 2: Different size of LLMs**:
> We were unable to locate the Llama-3.1 model with 1B parameters. However, detailed experiments are provided in the **Supplementary Material**, under the sections "Log-Loss for Classification" and "Regression Data." These experiments utilize three models: GPT-Neo with 1.3B parameters, Phi-3.5 with 3.5B parameters, and Llama-3.1 with 8B parameters. The results indicate that larger models exhibit more pronounced monotonicity and are more likely to have performance flattens in their final layers.
>
> **Weakness 3\& 4 and Question 1: using other metrics**:
> We have presented more experiments with different metrics and datasets and detailed information is introduced in the **Supplementary Material**. The results demonstrate that the monotonicity principle remains effective across various metrics and datasets. The label distribution is uniform. For example, there are 5 labels in the Yale dataset, then we randomly select 10000 samples and the sample size of each label is 2000.
>
> We hope these clarifications and the additional experiments address your concerns. Thank you for your thoughtful feedback and your review. Let us know if you have further questions or suggestions.

---

### Official Review · Reviewer_tNpt · 2024-11-19

**Soundness:** 2
**Presentation:** 2
**Contribution:** 2
**Rating:** 5
**Confidence:** 4

**Summary:**

In this paper, a hierarchical analysis of LLMs is presented to illustrate the phenomenon of monotonicity in layerwise performance. The study shows that this progressive improvement in test accuracy, observed across successive layer in pre-training data and generalizes to a variety of downstream tasks.

**Strengths:**

1. Many state-of-the-art LLMs are compared in experiments and many datasets from diverse domains are used in evaluation.
2. The experiments are reasonable and the results are convincing.

**Weaknesses:**

1. It'd be great if more experiments on the application of monotonicity could be conducted, for example, using more metrics other than test accuracy, and including more datasets.

**Questions:**

How does the size of training data affect the monotonicity of layerwise performance?

---

> ### Author Response · Authors · 2024-11-22
>
> Thank you for your positive feedback on our manuscript. We truly appreciate your recognition of the impact of our work.
>
> **Weakness: using other metrics**: We have presented more experiments with different metrics and datasets in the **Supplementary Material**. The results demonstrate that the monotonicity principle remains effective across various metrics and datasets.
>
> **Questions: the effect of increasing training data**: We have presented more experiments with training sample sizes, shown in the **Supplementary Material**, Monotonicity when increasing training samples. With a smaller dataset, the monotonicity property may not be immediately apparent due to the limited number of samples, which can introduce variability and noise into the results. However, as the sample size increases, the underlying patterns become more pronounced, and the monotonicity becomes more evident.
>
> We hope these additional experiments and results can address your concerns. Thank you again for your valuable feedback and consideration. Let us know if you have any questions or suggestions.

---

### Author Response · Authors · 2024-11-22
**Further Experiments Addressing Reviewers' Concerns**

Thank you for your thoughtful suggestions and insightful comments. Many reviewers expressed interest in exploring additional metrics and experiments, and we are pleased to provide further clarifications and results in this response. The following experiments and their results are detailed in the **Supplementary Material**:

**Perplexity in GPT-2**:
We evaluated the performance of GPT-2 using Perplexity on the OpenWebText dataset. The results demonstrate that the monotonicity principle remains effective across various metrics, even after the training process.

**Log-Loss for Classification**:
To assess classification performance, we calculated the log-loss on the Banking77 dataset across three models. The results indicate that the monotonicity property is consistently upheld across these models, even with this metric.

**MMLU Question Answering - Humanities**:
We conducted experiments on the MMLU Humanities question-answering benchmark to evaluate two models. The findings confirm that, while no accuracy flattens is observed in the final layers due to the complexity of this dataset—limiting the proposed method's ability to accelerate the inference process—the monotonicity principle remains valid for this task.

**Regression Data**:
For regression tasks, we evaluated the Yelp dataset for rating prediction (\(Y = \{1, 2, 3, 4, 5\}\)) using Mean Squared Error (MSE) as the evaluation metric. The results confirm that the monotonicity property is consistently maintained across all three models tested on this regression dataset.

We hope these additional experiments and results effectively address the reviewers' concerns and further demonstrate the robustness of our approach across various tasks and evaluation metrics. Thank you again for your valuable feedback and consideration.

---

### Author Response · Authors · 2024-11-23
**Follow-Up on Reviewer Responses**

Dear ACs and Reviewers,

I hope this message finds you well. I am writing to inquire about the next steps for our submission, for which we have carefully addressed all of the reviewers' concerns in our submitted rebuttals.

We sincerely appreciate your time and support throughout this process. Awaiting your guidance, we stand ready to implement any recommendations you may have.

Best regards

---

### Author Response · Authors · 2024-11-26

Dear ACs and Reviewers,

I hope this message finds you well. We kindly ask the reviewers to provide their comments on the authors' reply to the initial feedback, focusing on whether the responses adequately address the concerns raised and improve the manuscript.

If the reviewers believe that additional experiments would be beneficial to address any outstanding issues, we encourage them to request these experiments now, as November 26th is the final date for authors to upload a revised PDF with any new results. After this date, authors will only be able to post replies on the forum, and no further experiments can be added.

Your timely feedback and suggestions are crucial for ensuring the manuscript reaches its highest potential within the given timeline. We sincerely appreciate your time and support throughout this process.

Best regards

---

### Meta-Review · Area_Chair_9ao9 · 2024-12-18

**Metareview:**

This paper argues that LLM layerwise performance is "monotonic", i.e., deeper layers yield more accurate performance on downstream tasks. This is intuitive, given that deeper networks typically offer stronger performance, it would seem likely to follow that deeper layers within networks would similarly constitute stronger representations. This is a point made by both xvcr and XUzJ. Still, it is valuable to establish this empirically, even if not surprising. That said, the work does not engage adequately with existing related research, as pointed out by XUzJ.

**Additional Comments On Reviewer Discussion:**

The authors responded to critiques with supplementary results supporting some of their arguments. They also had an exchange with XUzJ, but I think failed to sufficiently address the main concerns that this reviewer raised.

---

### Decision · Program_Chairs · 2025-01-22

Reject